# Microwave imaging of quasi-periodic pulsations at flare current sheet

Yuankun Kou [1,2], Xin Cheng [1,2,3] ✉, Yulei Wang[1,2], Sijie Yu [4] ✉, Bin Chen [4], Eduard P. Kontar [5] & Mingde Ding [1,2]

Quasi-periodic pulsations (QPPs) are frequently detected in solar and stellar flares, but the underlying physical mechanisms are still to be ascertained. Here, we show microwave QPPs during a solar flare originating from quasi-periodic magnetic reconnection at the flare current sheet. They appear as two vertically detached but closely related sources with the brighter ones located at flare loops and the weaker ones along the stretched current sheet. Although the brightness temperatures of the two microwave sources differ greatly, they vary in phase with periods of about 10–20 s and 30–60 s. The gyrosynchrotron-dominated microwave spectra also present a quasi-periodic soft-hard-soft evolution. These results suggest that relevant high-energy electrons are accelerated by quasi-periodic reconnection, likely arising from the modulation of magnetic islands within the current sheet as validated by a 2.5-dimensional magnetohydrodynamic simulation.

Quasi-periodic pulsations (QPPs), the periodic variation of electromagnetic emissions, have often been observed in celestial environments of different scales ranging from flaring stars to cosmic-scale gamma ray and fast radio bursts. In the past decades, flare QPPs from the closest star, the Sun, have been investigated in detail, with the first one reported in 1960s[1,2]. Subsequently, they are proved to be prevalent at nearly all wavelength bands from radio to $\gamma$-rays. Their periods are found to be distributed in a wide range from sub-second to several minutes[3].

The mechanisms generating solar flare QPPs can be generally categorized into two groups: magnetohydrodynamic (MHD) waves modulated and oscillatory magnetic reconnection models[3–7]. The former explanation attributes flare QPPs to MHD waves within/among coronal loops, which can periodically modulate parameters of loops and then result in oscillatory emissions. The induced sausage mode[8–10] and kink mode[9,11,12], as well as the slow magnetoacoustic mode[13–15], are common candidates adopted to interpret observed QPPs. The corresponding characteristic period ranges from seconds, minutes to tens of minutes, respectively[7].

On the other hand, the oscillatory reconnection model argues that magnetic reconnection plays an essential role in driving most periodic processes of solar flares[16–19]. In the standard flare (CSHKP) model[20–25], magnetic reconnection takes place in a vertical current sheet (CS), which connects flare loops and erupting coronal mass ejections (CMEs), and accelerates electrons. Once the reconnection process within the CS presents an oscillatory behavior, which could be either spontaneous or driven by external MHD waves, electrons will be accelerated quasi-periodically. As a result, quasi-periodic microwave emissions are generated when accelerated electrons propagate along magnetic loops. Almost simultaneously, quasi-periodic hard X-ray (HXR) radiations also present as the electrons reach at footpoints of the loops, which also explains the similar temporal evolution pattern between HXR and microwave emissions[26]. Moreover, soft X-ray (SXR) and extreme-ultraviolet (EUV) emissions in flare loops radiated by hot plasma from electrons bombarding the chromosphere could also show a quasi-periodicity. Therefore, in the oscillatory reconnection scenario, quasi-periodicity of flare emissions at all bands is essentially caused by the quasi-periodic reconnection process[7].

[1]School of Astronomy and Space Science, Nanjing University, Nanjing 210093, China. [2]Key Laboratory of Modern Astronomy and Astrophysics (Nanjing University), Ministry of Education, Nanjing 210093, China. [3]Max Planck Institute for Solar System Research, Göttingen 37077, Germany. [4]Center for Solar-Terrestrial Research, New Jersey Institute of Technology, 323 Martin Luther King Jr. Blvd., Newark, NJ 07102-1982, USA. [5]School of Physics & Astronomy, University of Glasgow, Glasgow G12 8QQ, UK. ✉e-mail: xincheng@nju.edu.cn; sijie.yu@njit.edu

The microwave and HXR images provide the key information for diagnosing mechanisms of flare QPPs[16,27-33]. The first imaging observation of flare QPPs with the Hard X-ray Telescope onboard Yohkoh and the Nobeyama Radioheliograph (NoRH) revealed that accelerated electrons generate quasi-periodic microwave and HXR emissions[27]. With the NoRH data, Melnikov et al.[28] found microwave QPPs at different parts of flare loops and attributed them to the modulation of two oscillation modes in flare loops. Huang et al.[16] even clearly resolved the spatial distribution of the non-thermal microwave QPPs along a limb flare loop. However, due to the limited frequency coverage (at 17 GHz and 34 GHz), NoRH is almost impossible to observe the structure of the key energy release region (reconnection CS) above the flare loops, which is more sensitive to frequencies lower than 10 GHz[34], and is thus difficult to distinguish the role of quasi-periodic magnetic reconnection. Besides, the recent instrument Siberian Radioheliograph and Mingantu Spectral Radioheliograph also detected microwave QPPs, respectively, but both of which were found to be not directly related with the main energy release site[35,36].

The microwave radioheliograph Expanded Owens Valley Solar Array (EOVSA) in use since early 2017, with the state-of-the-art spectro-temporal resolution capability in microwave, has demonstrated a powerful potentiality in diagnosing energetic electrons and associated reconnection process[34,37-39]. At the time of the observation, it covers a frequency range of 3–18 GHz with 126 spectral channels combining into 30 spectral windows (SPWs). At each SPW, high-resolution images can be derived with a high cadence (up to 1 s). Such a spectral-imaging capability is desired to construct microwave images of the CS at multiple SPWs and extract broad-frequency-range spectra at each pixel. It has been documented by recent observations on the famous 2017 September 10 event, in which the microwave morphology of the flare loops and the extended CS can be well reconstructed at frequencies below 10 GHz[37,38]. The spatially resolved microwave spectra obtained at different spatial locations in the CS are used to derive the magnetic field strength and its distribution along the CS[34,40].

Here, we show microwave imaging of QPPs within the reconnection current sheet during a C5.9-class solar flare. Simultaneous observations of EOVSA and the Atmospheric Imaging Assembly (AIA) aboard the Solar Dynamics Observatory (SDO) reveal that the flare produces two vertically detached but closely related microwave sources at all wavelength bands from 3.4 to 10.9 GHz, which are located at the flare loops and extended along a stretched reconnection CS, respectively. Interestingly, the brightness temperatures and spectral indices of the microwave spectra in the two sources vary in phase and quasi-periodically. The 2.5-dimensional MHD simulation suggests the modulation of magnetic islands within the CS leads to the quasi-periodic magnetic reconnection and electron acceleration, which then generate microwave QPPs.

## Results
### Event overview
The flare on 2017 July 13 is located at active region 12667. It starts at about 21:46 universal time (UT) and peaks at about 21:55 UT (Fig. 1a). The EOVSA cross-power dynamic spectrum (DS) in the frequency range of 3.4–17.9 GHz clearly shows that some microwave bursts appears repetitively during the energy release process of the flare. The quasi-periodic emission extends to the frequency of 18 GHz near the flare peak time (Fig. 1b). The metric DS observed by the radio spectrometer GREENLAND of the Compound Astronomical Low frequency Low cost Instrument for Spectroscopy and Transportable Observatory (e-CALLISTO/GREENLAND) also presents a group of type III bursts (Fig. 1c). These type III bursts that are believed to be caused by escaped electrons along the open field seem to also appear quasi-periodically, well accompanying the microwave bursts, in particular before 21:56 UT. Such a synchronization implies that the bursts at both wavelengths

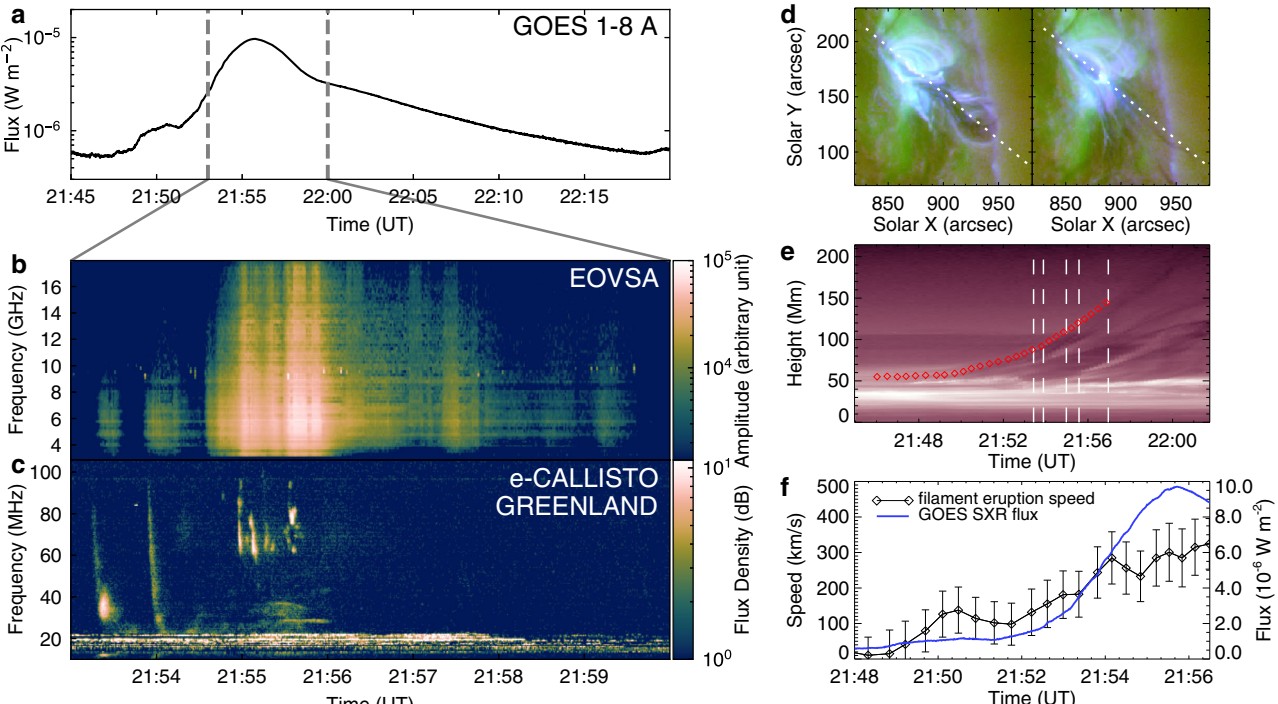

**Fig. 1 | Overview of the flare event. a** GOES SXR 1–8 Å flux curve of the flare. **b**, **c** EOVSA cross-power and e-CALLISTO/GREENLAND DS during 21:53–22:00 UT. **d** SDO/AIA composite images of 131 (blue), 171 (red) and 211 Å (green) at 21:54:58 UT and 22:09:22 UT. The white dotted lines represent slits along the direction of the filament eruption. **e** AIA 211 Å slice-time plot showing the evolution of the filament top. The diamond symbols in red represent the measured heights, and the white dash lines indicate the five instants of snapshots in Fig. 2. **f** Temporal evolution of the filament speed (black) and the GOES SXR 1–8 Å flux (blue) of the associated flare. The speed errors are from the uncertainty in height measurements (5 pixels, about 2.14 Mm).

are most likely of the same origin, presumably generated by flare-accelerated electrons.

EUV images recorded from the flare impulsive to decay phase show that the flare is caused by the eruption of a filament (Fig. 1d). At about 21:48 UT, the filament starts to rise, giving rise to EUV brightenings near its two footpoints simultaneously. Afterwards, the filament quickly erupts and the SXR emission rapidly increases. Nevertheless, due to the obscuration of the filament, the eruption-induced cusp-shaped loops are not observed until after about 21:57 UT. During the main phase, only two curved flare ribbons are visible and located at both sides of the remaining filament. After about 23:12 UT, the associated CME appears in the field of view of the Large Angle and Spectrometric Coronagraph onboard Solar and Heliospheric Observatory.

We measure the height of the erupting filament (red diamonds in Fig. 1e) and calculate the eruption speed through the first derivative of the height-time data (Fig. 1f). We find that the speed varies in phase with the Geostationary Operational Environmental Satellites (GOES) SXR 1–8 Å flux of the flare. With the SXR flux rising rapidly at about 21:52 UT, the filament speed also quickly increases, reaching to about 300 km s$^{-1}$ at the flare peak time. The average acceleration is about 440 m s$^{-2}$. During about 21:48 UT–21:52 UT, the filament also shows a period of acceleration, but the SXR emission only increases slightly, probably due to the obscuration of the filament.

## Microwave and EUV imaging

Thanks to EOVSA's spectral-imaging capability at a high cadence (1 s), the dynamic evolution of the microwave emission sources during the flare can be determined with a high precision. The 4.4 GHz emission consists of a primary strong source and a secondary weak source, with the latter connecting to the main one and extends along the eruption direction (Fig. 2b). These sources seem to enhance repeatedly but their morphology is nearly unchanged. To highlight the secondary weak source, we further fit the main source with a two-dimensional Gaussian model and then subtract the fitted source from the observed images. The contours of the main and second sources clearly show that the double-source feature persistently exists at all observed frequencies (Fig. 2c and Supplementary Movie 1).

A comparison of the microwave sources with the AIA 211 Å images clearly displays that the strong sources are located at the flare loops and their centroids at different frequencies are very close to each other. For the weak sources, however, their centroids ascend along the direction of the eruption with decreasing frequency (Fig. 2c). These relatively weak microwave sources closely resemble the observations previously reported in the much stronger 2017 September 10 X8.2 flare[34] and are most likely caused by non-thermal electrons distributed along a long reconnection CS stretched by the erupting filament as predicted in the standard model (Fig. 2d). This is further supported by the presence of reconnection downflows along the CS, which appear intermittently above and quickly move toward the top of flare loops after the filament eruption (Supplementary Fig. 1), consistent with previous argumentation[41–43]. Considering that the average magnetic field of the CS decreases with height above the primary X point[34], the frequency-dependent distribution of the weak microwave sources observed here complies with the picture that the peak frequency of the gyrosynchrotron (GS) emission decreases with a decreasing magnetic field strength[44,45].

## Quasi-periodicity of microwave sources

With spatially resolved microwave images, we study the temporal variation of the brightness temperature, $T_b(t, v)$, at each frequency $v$. The $T_b(t)$ of the two sources, both at an optically-thick frequency 3.4 GHz and an optically-thin frequency 8.4 GHz, increases and decreases repeatedly and in phase, presenting an evident quasi-periodicity (Fig. 3a, b) and high relation to each other. Furthermore, the temporal evolution of $T_b(t)$ is also found to be approximately

synchronous with that of the time derivative of SXR flux, with a time lag of about 13 s (Supplementary Fig. 2). Wavelet analysis provides the periodicity of these curves. For both the strong and weak microwave sources at 8.4 GHz, a confident period appears in the range of about 10–20 s between about 21:55 UT and about 21:56 UT; while the period of about 30–60 s appears confidently only in the weak sources from about 21:54 to 21:57 UT (Fig. 3c–e). Similarly, the time derivative of SXR flux also presents a period of about 10–20 s around 21:54 UT as well as between about 21:55 UT and about 21:56 UT. The synchronization and similar periodicity indicate that the Neupert effect, referring to the similarity of the temporal evolution between the microwave/HXR flux and the time-derivative of SXR flux of flares, applies to this event. It shows that non-thermal electrons producing the HXR and microwave emissions are the primary heating source of the flare loops responsible for the SXR emissions[46,47].

## Spectral properties of microwave sources

The brightness temperature spectra of the strong and weak but long extended microwave sources, the latter of which is divided into three small regions along the eruption direction and labeled as Region 0, 1, and 2 in Fig. 4c, both show a power-law form with a positive slope at lower frequencies and a negative slope at higher frequencies (Fig. 4a, b). This is a characteristic of GS emission spectra. Using the power-law model, we fit the optically-thin part of the spectra (from the peak frequency to 10.9 GHz, above which the signal-to-noise ratio becomes too low) and derive the (photon) spectral indices as shown in Fig. 4d. It is found that these spectral indices are always larger than the typical value for the optically-thin parts of thermal GS spectra (-10)[44,45], which indicates that the microwave spectra observed here are most likely of a non-thermal origin.

We also compare qualitatively the non-thermal properties at different time and different regions. The spectral indices for microwave sources at the flare loops and the lower parts of the expected CS (Region 0 and 1) are found to be largely synchronized with the microwave brightness temperature of the CS sources at the optically-thin frequency band 8.4 GHz and exhibit a repeated soft-hard-soft variation behaviour (Fig. 4d and Supplementary Fig. 3). Comparing with Region 0 and 1, the temporal evolution of the spectral index for Region 2 seems to be similar but with a delay of about half a minute, in particular after about 21:54:30 UT (Fig. 4d). The repeated soft-hard-soft feature detected here implies that electrons responsible for the microwave sources are accelerated quasi-periodically during the flare. Moreover, the spectral indices at the different regions are evidently distinct. They are the hardest (less negative) at Region 0, significantly decrease (softer) toward Region 1 and 2, but slightly decrease (softer) at the flare loops. This could be due to that the lowest part of the CS (Region 0) is close to the reconnection termination shock, where electrons trapped in the magnetic mirror are accelerated most efficiently. The acceleration is relatively less efficient in the CS above the termination shock, even further weakening with the increase of the height such as from Region 0 to 2 (also see ref. 34, 40, 48). While for the microwave spectra at the flare loops, they are likely generated by escaped electrons from the acceleration region above the flare loop-top. Together with the fact that electrons with lower energies escape more easily[49], the spectral indices are thus smaller (softer) than that at Region 0.

## MHD simulation of quasi-periodic magnetic reconnection

To explore the origin of quasi-periodic energy release during the flare reconnection, we run a high-resolution 2.5-dimensional resistive MHD simulation. We find that, during the fast reconnection, magnetic islands are quasi-periodically generated near X-points in the main CS. The islands move downwards and then interact with the flare loops quasi-periodically (Fig. 5a–c). Owing to the modulation of the islands, the energy release (or reconnection) rate presents a quasi-periodicity

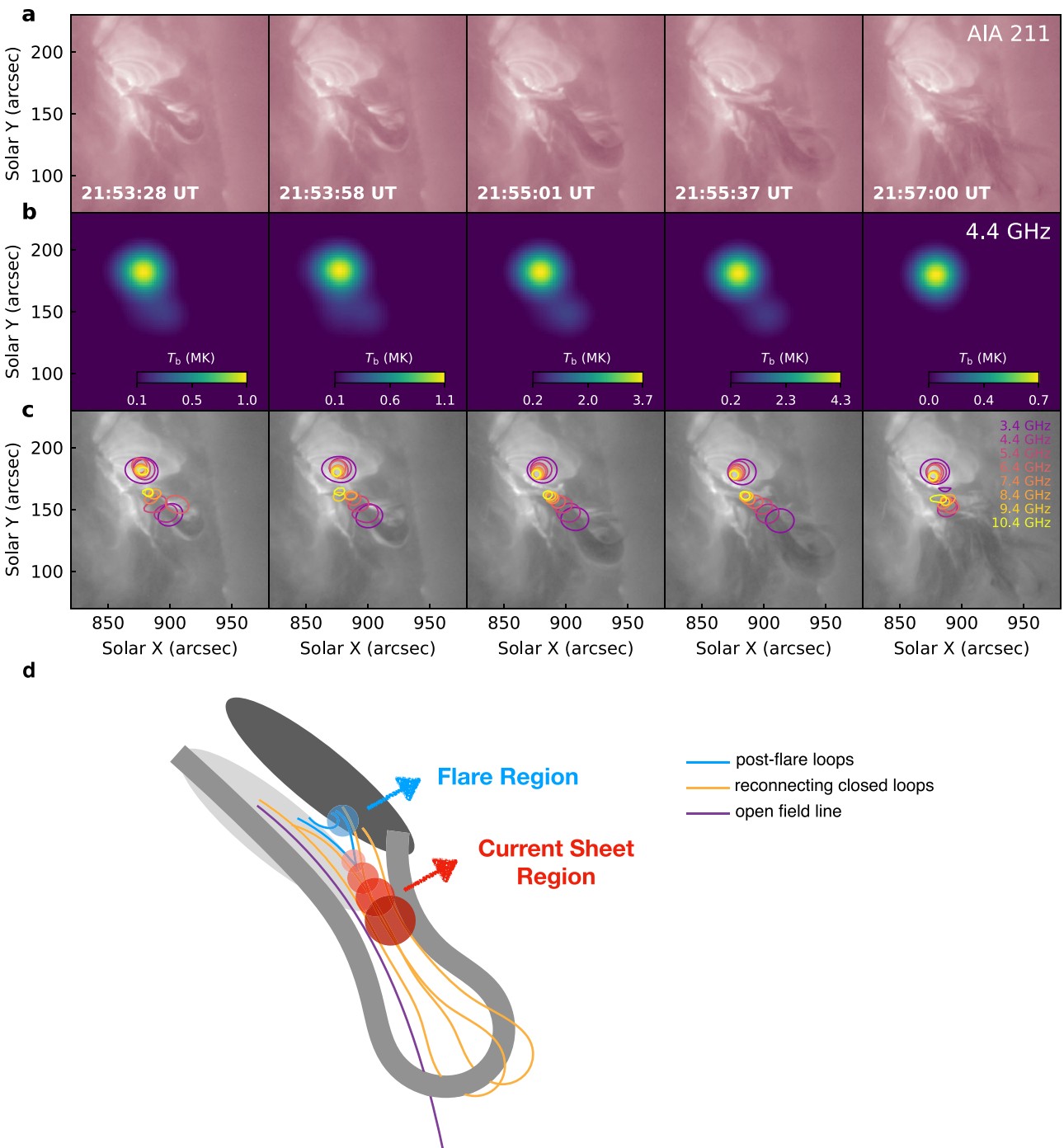

**Fig. 2 | EUV and microwave emissions of the flare. a** SDO/AIA 211 Å images showing the erupting filament and induced EUV brightenings. **b** EOVSA microwave images displaying the distribution and evolution of the flare brightness temperature, $T_b$, at 4.4 GHz. **c** SDO/AIA 211 Å images with superimposed microwave contours. The color from purple to yellow correspond to frequencies from 3.4 GHz to 10.4 GHz. There are two groups of contours, each of which shows 80% of the highest $T_b$ in the strong and weak sources, respectively. **d** A schematic diagram interpreting the magnetic field structure during the eruption. The wide curve in grey represents the erupting filament with the overlying field in gold and purple. The ovals in shallower and deeper grey represent magnetic positive and negative polarities, respectively. The circles in blue and red, from shallower to deeper with decreasing frequency, indicate the strong and weak microwave sources locating at the flare loops and reconnection CS region, respectively.

(see Fig. 2 of ref. 50). Such a quasi-periodic energy release process may accelerate electrons quasi-periodically, thus generating the QPPs at the microwave bands that we have observed. In fact, the quasi-periodic characteristics related to the generation of islands in the CS has been revealed by numerous previous MHD simulations[51–55]. It is expected that waves and shocks generated near the ends of the CS by (even quasi-steady) reconnection outflows[56,57] or the quasi-periodic deformation of the CS structure[58–60] could also lead to oscillations of various flare structures and emissions.

Based on the simulated density and temperature distributions, we also calculate the synthetic thermal X-ray emission. We find that, starting from $t = 300$ s, the SXR emission power at 1–8 Å ($P_{SXR}$) first grows slowly, then rises rapidly during $t = 500$–550 s, and reaches its peak near $t = 575$ s (Fig. 6a). Both the curves of $P_{SXR}$ and its time

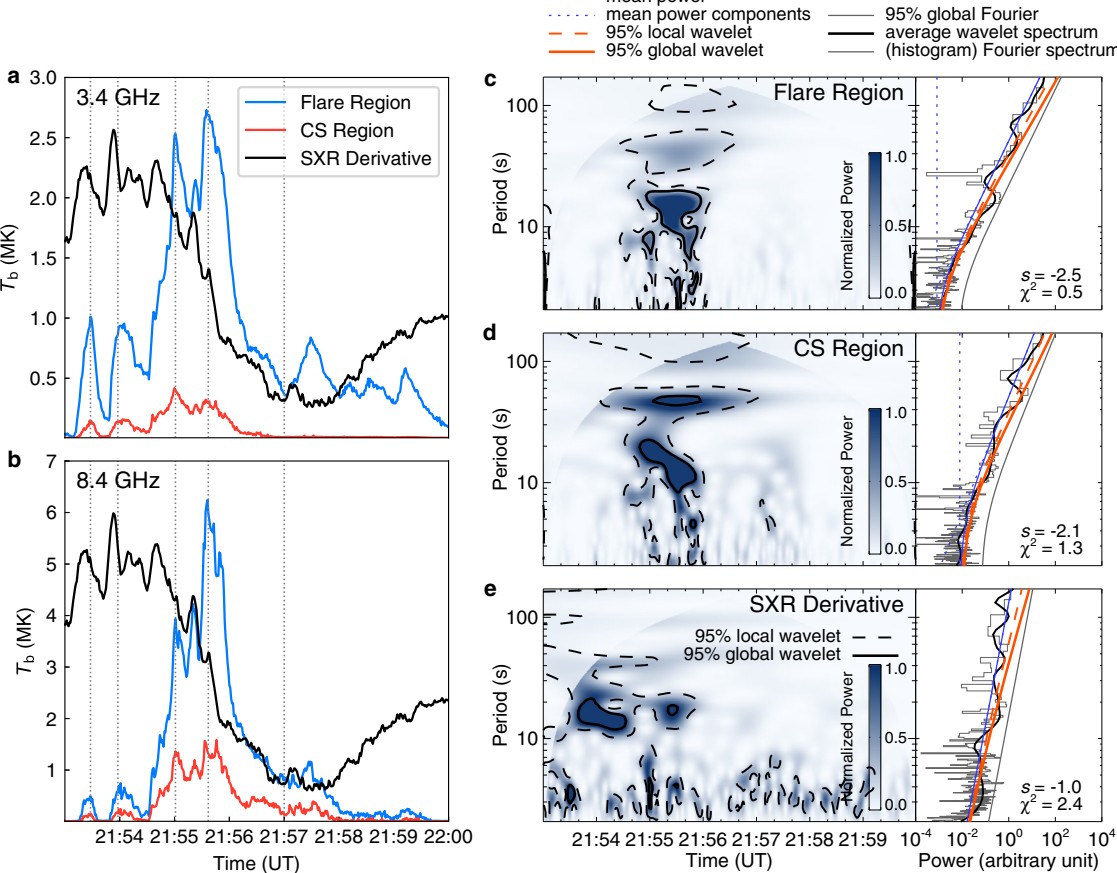

**Fig. 3 | Temporal variations and periodicity of brightness temperature and time derivative of SXR flux.** **a** The variations of brightness temperature, $T_b$, the maximal values within corresponding boxes as in Fig. 4c, of the microwave sources at the flare loops (blue, Region C) and the entire CS region (red, including Region 0, 1, and 2) at the frequency of 3.4 GHz. The black curves represent the 11-second-smoothed time derivative of the GOES SXR 1–8 Å emission. The vertical dot lines mark the five instants when the resolved microwave and EUV images are shown in Fig. 2. **b** The same as (**a**) but at 8.4 GHz. **c** The normalized wavelet power spectrum of $T_b$ variation of the flare loops region at 8.4 GHz and time-averaged Fourier/ wavelet spectrum. The region with shallower color denotes the cone of influence (COI), under which the periods are insignificant. The black dotted contours enclose

the regions with local wavelet significance higher than 95%, and the solid contours enclose the regions with global wavelet significance higher than 95%. Fitting results of the Fourier spectrum (histogram) are plotted with the blue solid line, with mean power components in blue dotted lines. The thick black line is the average wavelet spectrum. The gray solid line marks the 95% significance level of the Fourier spectrum, the red dotted line the 95% local time-averaged significance level, and the red solid line the 95% global time-averaged significance level. Fitted power-law index ($s$, see Formula (1)) and the least chi-square value ($\chi^2$) are shown in the lower right corner of the panel. **d**, **e** The same as (**c**) but for the wavelet analysis of the $T_b$ variation in the entire CS region at 8.4 GHz, and of the time derivative of SXR flux of the flare, respectively.

---

derivative show quasi-periodic features similar to the GOES SXR lightcurve and its time derivative, respectively (Fig. 3 and Fig. 6a–c). Similar to the observed periodicity, the wavelet analysis of $dP_{SXR}/dt$ also presents two primary periodic components, with the stronger one of about 16–30 s and the weaker other of about 30–60 s (Fig. 6c). Especially, the number distribution of the magnetic islands as a function of magnetic flux and time in the integration region shows that the formation of the QPPs is closely related to the generation of magnetic islands (Fig. 6d). In short, the comparability between sythetic and observed periodicity supports that the oscillatory magnetic reconnection is the main cause of the QPPs as detected for the current event.

## Discussion

In this work, we analyze an eruptive flare on 2017 July 13 focusing on its spatially resolved microwave emissions observed by EOVSA. Combining EUV images provided by SDO/AIA, we find that the microwave emissions are mainly from two vertically detached but closely related sources. The main strong sources are located at the flare loops with their centroids almost unchanged at different frequencies. For the second weak ones, they extend toward the direction of the filament eruption with their centroids ascending with decreasing frequency.

This provides strong evidence that the microwave sources are emitted by non-thermal electrons accelerated in the close vicinity of the flare CS as found by recent observations[34,37,38,40,42].

The most striking finding is that the brightness temperature and spectra for the two CS-associated microwave sources vary synchronously and present an obvious quasi-periodicity. Through wavelet analysis, we find that the main period is in the range of about 10–20 s and the second period in the range of about 30–60 s. Such periodicity is also observed in the time derivative of the spatially integrated SXR emission. The periods observed here are basically compatible with previous results obtained in other events and at different wavelengths[17,61]. Moreover, the similarity between the evolution of the microwave brightness temperature and the time derivative of SXR flux hints the major role of non-thermal electrons in heating the flare plasma and exciting the corresponding emissions. We also find that the optically-thin parts of the spectra tend to be flatter (harder) than those for the typical thermal GS emission[45]. We suggest that the quasi-periodic flare reconnection during the event drives the acceleration of the non-thermal electrons and then produces quasi-periodic microwave emissions at both the reconnection CS and the flare loops. Our 2.5-dimensional

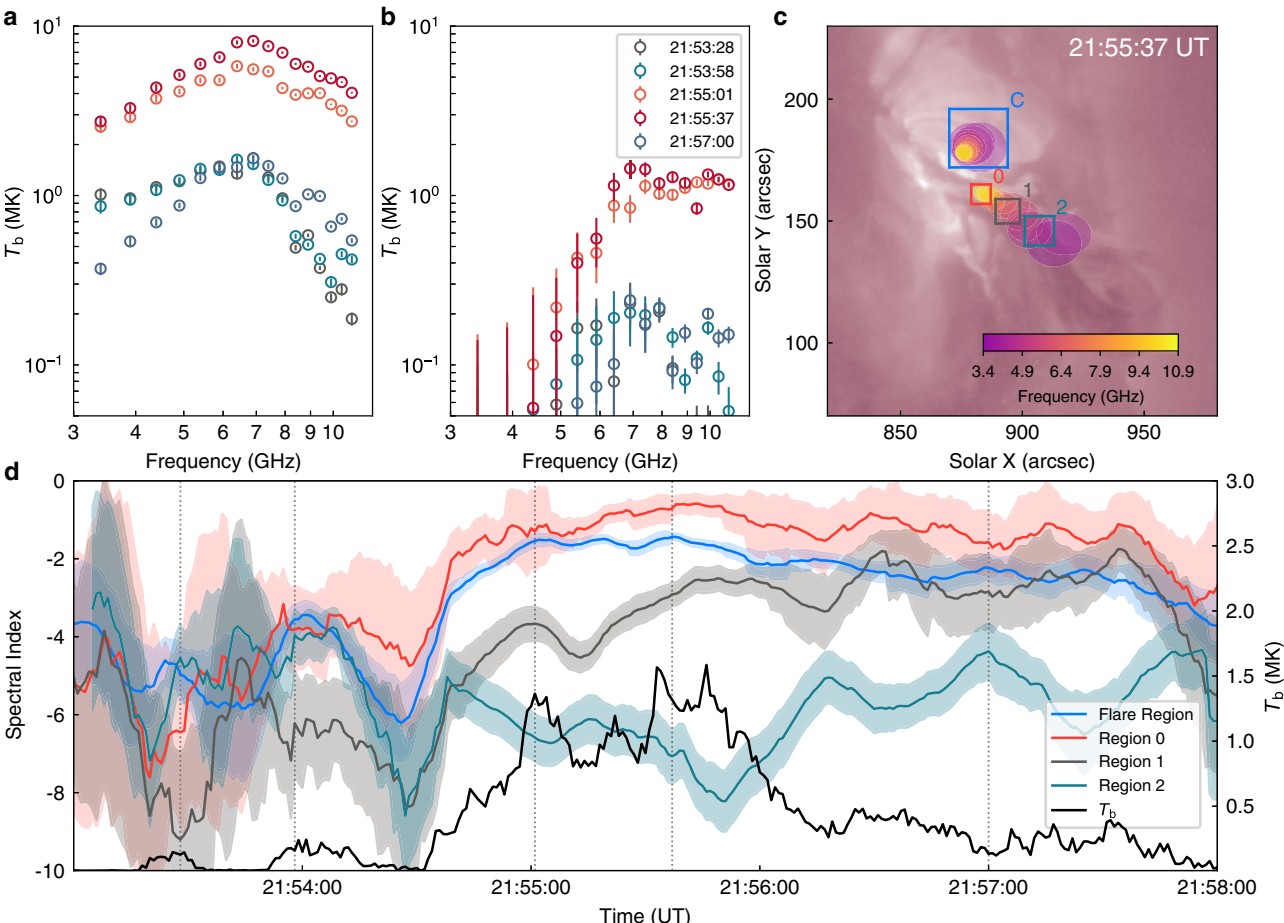

**Fig. 4 | Spatially resolved microwave spectra and their temporal evolution.** Microwave spectra for the flare loops (Region C; **a**) and the lower part of the CS (Region 0; **b**) in the frequency range of 3.4–10.9 GHz. The brightness temperatures, $T_b$, refer to maximal values in the corresponding boxes as shown in panel (**c**). The errors are the root-mean-squares of residual images, which are derived by subtracting two-dimensional Gaussian fitted microwave sources from the observed ones. **c** The AIA 211 Å image overlaid by the contours of microwave sources at

21:55:37 UT. Boxes with different colors show the regions for deriving the microwave spectra. **d** Temporal evolution of spectral indices (11-second-smoothed) in the frequency range from the peak to 10.9 GHz for the flare loops (Region C) and the stretched reconnection region (Region 0, 1 and 2), with the errors indicated by translucent regions. The errors are the standard deviations. The black curve represents the variation of the maximal $T_b$ in the entire CS region at the frequency of 8.4 GHz, and the vertical dot lines the five instants shown in Fig. 2a.

MHD numerical simulation interprets that the quasi-periodicity of the magnetic reconnection is caused by the appearance of magnetic islands within the long-stretched CS, which is further testified by the similarity between the periods of the synthetic and observed SXR emission. It is also possible that the quasi-periodic reconnection is caused by an external driver that quasi-periodically changes the reconnection inflows[3–7], which, however, is hardly detected in the current event.

The periodicity of the QPPs we derived seems to support the interpretation of the sausage mode, which is also able to generate flare QPPs at the microwave bands via modulating the magnetic field strength and the electron density within flare loops[3]. Theoretically, the sausage mode can produce expansion/shrinkage of flare loops[8], but which is not observable for this event even with EUV images of the high spatio-temporal resolution. Moreover, the QPP phase difference between high (optically-thin bands) and low frequencies (optically-thick bands) predicted in the MHD model[62,63] is also absent (see Fig. 3a, b). Thus, we tend to exclude the possibility of sausage mode giving rise to the QPPs as detected along the CS. Of course, the sausage mode in the flare loops could leak out when its wavelength exceeds a cutoff value[8]. But this is essentially equivalent to external MHD waves giving rise to the quasi-periodic reconnection within the CS.

The spatially resolved EOVSA data clearly locate the sources of the flare QPPs, in particular the CS sources above the flare loops, which provides convincing evidence for quasi-periodic magnetic reconnection as a cause of quasi-periodic flare emissions. Due to the quasi-periodicity of the reconnection, electrons are accelerated quasi-periodically either in the reconnection CS or in the termination shock above the flare loop-top, later of which is driven by quasi-periodic plasma downflows[42,43], and produce quasi-periodic microwave emissions at the reconnection region. As the energetic electrons are injected quasi-periodically into the flare loops, they also excite quasi-periodic microwave emissions there. Moreover, the flare also produces a group of quasi-periodic type III bursts, the peaks of which coincide well with that of the microwave emissions. This supports the speculation that energetic electrons originating from the quasi-periodic reconnection also produce quasi-periodic type III bursts once they escape from the reconnection region and propagate away from the Sun.

The separation of the two microwave sources in the eruption direction is similar to but largely different from that of double HXR sources in the corona discovered by Ramaty High Energy Solar Spectroscopic Imager observations[64–66]. In the framework of the standard flare model, the two vertically separated HXR sources are interpreted as those formed by the interactions of two oppositely

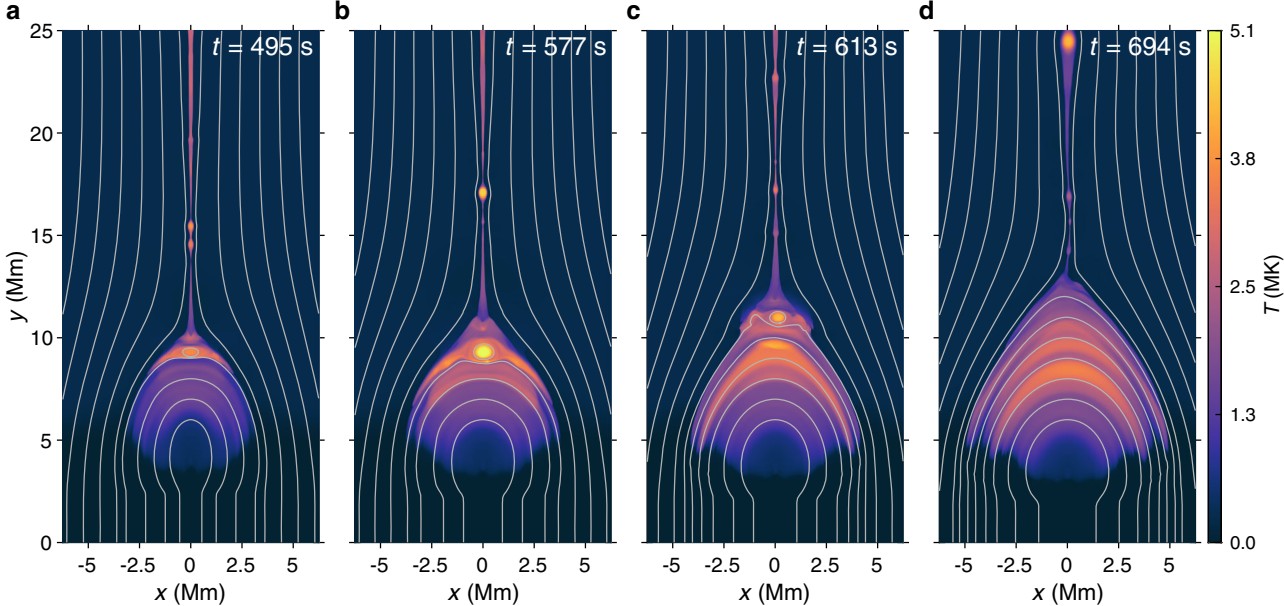

**Fig. 5 | Temperature distributions of the CS at four typical instants. a–c** The CS is fragmented into a number of magnetic islands. A relatively large magnetic island is interacting with the flare loop-top and several islands in the main CS are moving downwards. The gray curves mark the magnetic field lines. **d** The spatial scale of the flare loops becomes comparable with observations at later time.

directed reconnection outflows with flare loops and the erupting flux rope structure, respectively. The upper coronal HXR sources that rise with the eruption usually have softer spectra than the lower sources[65]. On the contrary, the microwave sources at the CS in our event do not significantly change their locations with time, and their lowest part has a harder spectral index than that of the flare-loop sources. These facts further support that the upper microwave sources are along the stretched reconnection CS with the lower tip probably corresponding to the termination shock as argued previously[48].

## Methods

### Microwave data reduction and spectral analysis

At the time of the observation, EOVSA has 126 spectral channels covering a frequency range of about 3–18 GHz, which combine into 30 SPWs to increase the dynamic range when imaging (so-called multi-frequency synthesis technique). The band width of each SPW is 500 MHz, and central frequencies are $f = 2.92 + n/2$ GHz, where $n$ varies from 1 to 30, respectively. The primary beam size of EOVSA is $186''.2/f$(GHz) $\times 35''.2/f$(GHz) at the observation time, and we adopt a circular restoring beam $102''.7/f$(GHz) to recover microwave images. We perform CLEAN iterations[67] for a time-sequence of images at SPWs over 3.4–10.9 GHz, until the peak residual $T_b$ down to a level of about $10^{-1}$ MK and thus the signal-to-noise ratio, in most cases, higher than 10. Tools for all these processes are included in the CASA (https://casa.nrao.edu) and SunCASA (https://github.com/suncasa/suncasa-src) packages.

High-resolution microwave images at multiple SPWs make it possible to obtain spatially-resolved microwave spectra at each time. We fit the optically-thin part of the spectra with the power-law model to get the spectral indices at different instants or regions. In practice, the optically-thin part is defined as the frequency range from the peak frequency, of each specific spectrum, to 10.9 GHz. For some spectra, if less than four frequency points remain within the optically-thin part (typically because of the relatively low spectra quality), they are not fitted. To reduce such spectra, an 11-second smoothing is applied.

### Wavelet analysis

We utilize the wavelet analysis to quantify the periodicity of the flare microwave emissions. The core wavelet software is provided by Torrence and Compo[68] and available at http://atoc.colorado.edu/research/wavelets; the code for fitting the background power spectra and determining the significance levels is adopted from ref. 69. Here, we use the mother wavelet 'Morlet' and fit the background Hann-window-apodized Fourier spectra with the combination of a power-law function and a constant value as

$$\sigma(\nu) = A\nu^s + C , \tag{1}$$

where $\sigma$ is the expected value of power and $\nu$ is frequency, to determine the 95% significance levels. We choose the 95% global significance level to confirm the periodicity, which takes into account the total number of degrees of freedom in the wavelet power spectra and is argued to be more reliable than the 95% local significance level traditionally used by ref. 68.

### MHD simulation

The numerical simulation in this work is the same as what introduced in ref. 50, in which the effects of anisotropic thermal conduction and chromospheric evaporation are included. The MHD equation we solve is

$$\frac{\partial \rho}{\partial t} + \nabla \cdot (\rho \mathbf{u}) = 0 ,$$

$$\frac{\partial (\rho \mathbf{u})}{\partial t} + \nabla \cdot \left( \rho \mathbf{u}\mathbf{u} - \mathbf{B}\mathbf{B} + P^*\mathbf{I} \right) = 0 ,$$

$$\frac{\partial e}{\partial t} + \nabla \cdot \left[ (e + P^*)\mathbf{u} - \mathbf{B}(\mathbf{B} \cdot \mathbf{u}) \right] = \nabla \cdot \left( \kappa_{\parallel} \hat{\mathbf{b}}\hat{\mathbf{b}} \cdot \nabla T \right) , \tag{2}$$

$$\frac{\partial \mathbf{B}}{\partial t} - \nabla \times (\mathbf{u} \times \mathbf{B}) = -\nabla \times (\eta \mathbf{J}) ,$$

$$\mathbf{J} = \nabla \times \mathbf{B} ,$$

where $P^* = p + B^2/2$, $e = p/(\gamma-1) + \rho u^2/2 + B^2/2$, $\kappa_{\parallel} = \kappa_0 T^{2.5}$, $\mathbf{I}$ is the identity matrix, and standard notations of variables are used. All

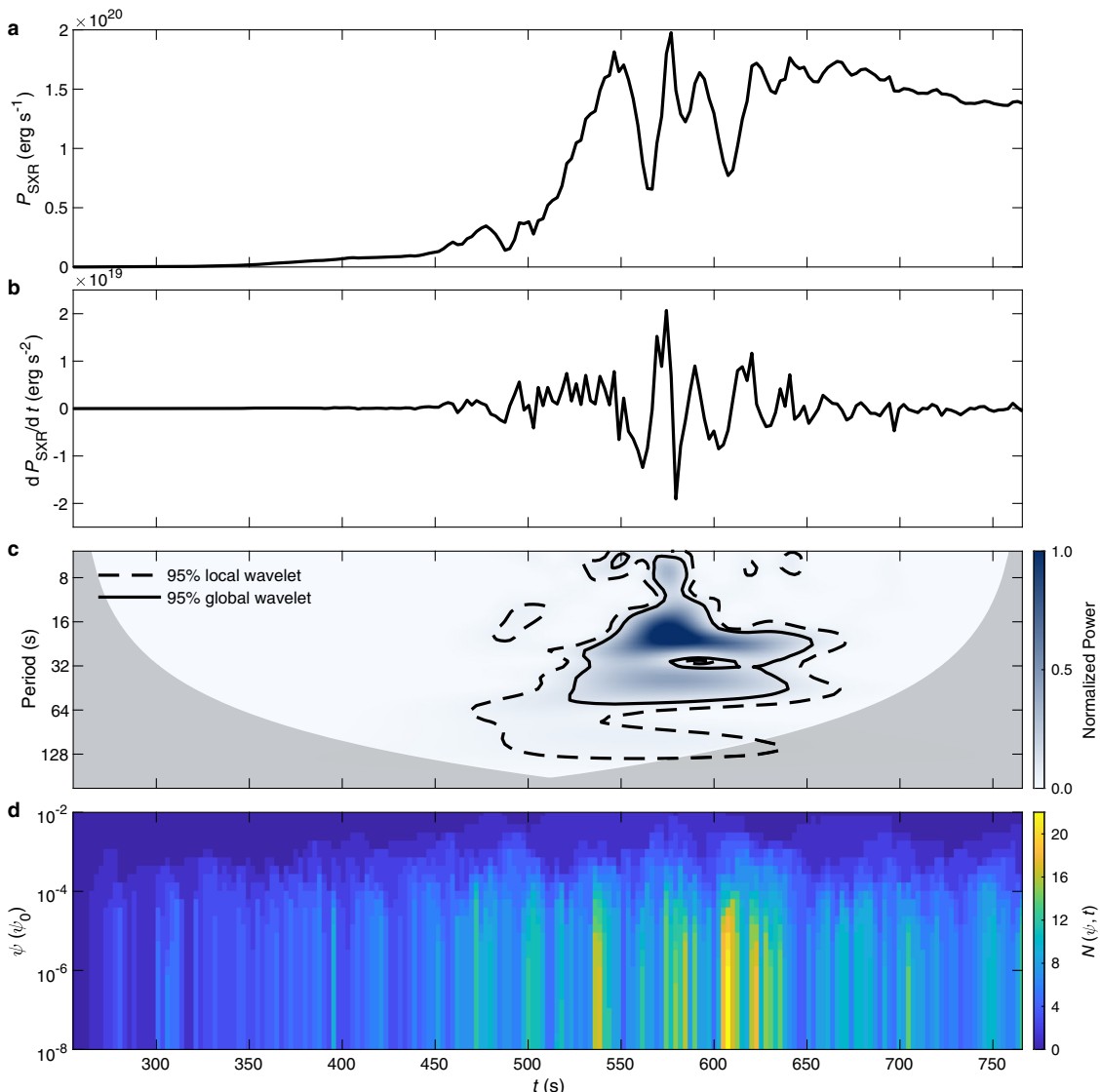

**Fig. 6 | Quasi-periodic characteristics of synthetic SXR variation. a** Temporal evolution of the SXR emission power at 1–8 Å, $P_{SXR}$. **b** Temporal evolution of the time derivative of $P_{SXR}$, d$P_{SXR}$/d$t$. **c** The wavelet power spectrum of the temporal evolution of d$P_{SXR}$/d$t$. Same as in Fig. 3, the gray shading denotes the cone of influence (COI), and the black contours enclose the regions where the global (solid line) and local (dashed line) significance is higher than 95%. **d** The evolution of the number distribution function of the magnetic island flux, $N(\psi,t)$, in the integration region, where $\psi$ is the flux of magnetic islands and $\psi_0 = 10^{11}$ Mx cm$^{-1}$. $N(\psi,t)$ is calculated by the same method as in ref. 50.

quantities are dimensionless in our simulation. We suppose an initial atmosphere with a uniform background pressure $p_b = 0.02$ and a density distribution

$$\rho(y) = \rho_{chr} + \frac{\rho_{cor} - \rho_{chr}}{2}\left[\tanh\left(\frac{y - h_{chr}}{l_{tr}} + 1\right)\right], \quad (3)$$

where $\rho_{cor} = 1$, $\rho_{chr} = 10^5$, $h_{chr} = 0.1$, and $l_{tr} = 0.02$. The simulation starts from a simplified CSHKP CS with a magnetic field profile

$$B_x = 0,$$
$$B_y = \begin{cases} \sin\left(\frac{\pi x}{2\lambda}\right), & |x| \leqslant \lambda, \\ 1, & x > \lambda, \\ -1, & x < -\lambda, \end{cases} \quad (4)$$
$$B_z = \sqrt{1 - B_x^2 - B_y^2},$$

where $\lambda = 0.1$ is the half-width of the CS. A localized anomalous resistivity is initially applied to trigger the fast reconnection, which is expressed by

$$\eta = \begin{cases} \eta_b + \eta_a \exp\left[-\frac{x^2 + (y - h_\eta)^2}{l_\eta^2}\right], & t \leq t_\eta, \\ \eta_b, & t > t_\eta, \end{cases} \quad (5)$$

where, $\eta_a = 5 \times 10^{-4}$, $h_\eta = 0.5$, $l_\eta = 0.03$, $t_\eta = 5$, and $\eta_b = 5 \times 10^{-6}$. The boundary condition is set as follows. The left ($x = -1$) and right ($x = 1$) are free boundaries, the top ($y = 4$) is a no-inflow boundary, and the bottom ($y = 0$) is a symmetric boundary. The above system is simulated with Athena 4.2. The conservation part of MHD equation is solved by the HLLD Riemann solver, the 3-order piece-wise parabolic flux reconstruction algorithm, and the Corner Transport Upwind (CTU) method. The resistivity and TC are calculated by the explicit operator splitting method. We set a high-precision uniform Cartesian mesh with 3840 and 7680 grids on $x$ and $y$ directions, respectively.

Here, we use new characteristic parameters for normalization based on the space and time scales of the observed flare in order to compare with results derived from observations. The characteristic

length, time, and magnetic field are set as $L_0 = 2.5 \times 10^9$ cm, $t_0 = 51$ s, and $B_0 = 40$ G, respectively. The characteristic velocity, namely the background Alfvén speed, is $u_0 = L_0/t_0 = 4.90 \times 10^7$ cm s$^{-1}$. Furthermore, we assume that the corona is composed of fully ionized hydrogen which gives the averaged particle mass $\bar{m} = 0.5 m_p = 8.36 \times 10^{-25}$ g, where $m_p$ is the mass of proton. The characteristic mass density can thus be derived as $\rho_0 = B_0^2/4\pi u_0^2 = 5.31 \times 10^{-14}$ g cm$^{-3}$, which corresponds to the electron density $n_{e0} = 0.5\rho_0/\bar{m} = 3.17 \times 10^{10}$ cm$^{-3}$. Then, the characteristic pressure, temperature, conductivity, and diffusivity can be respectively deduced as $p_0 = \rho_0 u_0^2 = 127.3$ dyn cm$^{-2}$, $T_0 = \bar{m} u_0^2/k_B = 14.52$ MK, $\kappa_{\parallel 0} = k_B \rho_0 u_0 L_0/\bar{m} = 1.07 \times 10^{12}$ erg s$^{-1}$ cm$^{-1}$ K$^{-1}$, and $\eta_0 = L_0 u_0 = 1.22 \times 10^{17}$ cm$^2$ s$^{-1}$, where $k_B$ denotes the Boltzmann constant. In this simulation, the ratio of specific heat is $\gamma = 5/3$ and the Spitzer conduction coefficient is set as $\kappa_0 = 10^{-6}$ erg s$^{-1}$ cm$^{-1}$ K$^{-3.5}$. Under this configuration, the total simulation time is 765 s and the duration of fast reconnection, starting at $t = 300$ s, is comparable with the duration of SXR peak emission, namely 7 min (21:53–22:00 UT), of the flare. Meanwhile, at the end of the simulation, the footpoint separation of the flare loops is about 10 Mm, which is also comparable with that of the real flare observed at AIA 131 Å after 22:00 UT (Fig. 1d and Fig. 5d).

The synthetic SXR power density in the frequency window $(\nu, \nu + d\nu)$ is evaluated by ref. 70 as

$$F_{SXR} = C T^{-1/2} \left( \sum_i n_i Z_i^2 \right) n_e \exp\left( -\frac{h\nu}{k_B T} \right) g_{ff}(h\nu, T), \quad (6)$$

where

$$C = \frac{2^5 \pi}{3 m_e c^3} \left( \frac{e}{\sqrt{4\pi\epsilon_0}} \right)^6 \left( \frac{2\pi}{3 k_B m_e} \right)^{1/2}, \quad (7)$$

and

$$g_{ff}(h\nu, T) = \begin{cases} 1, & h\nu \le k_B T, \\ \left( \frac{k_B T}{h\nu} \right)^{0.4}, & h\nu > k_B T, \end{cases} \quad (8)$$

where $e$ is the elementary charge, $m_e$ is the mass of electron, $c$ is the speed of light in vacuum, $h$ is the Planck constant, $\epsilon_0$ is the permittivity in vacuum, $n_e$ is the number density of electrons, and $n_i$ and $Z_i$ are the number density and charge number of ions of element $i$, respectively. Here, for simplicity, we assume that the plasma is composed of pure hydrogen. The X-ray emission from a two-dimensional region, enclosing the reconnection sites in the main CS and the flare loops, at the frequency window $[\nu_0, \nu_1]$ is thus calculated by

$$P_{SXR} = L_{LOS} \int_R dx dy \int_{\nu_0}^{\nu_1} \frac{F_{SXR}}{4\pi} d\nu, \quad (9)$$

where the integration region, $R$, is set as $x \in [-12.5, 12.5]$ Mm and $y \in [6.25, 25]$ Mm, and $L_{LOS} = 10^9$ cm is the estimated scale of the CS along the line of sight (LOS). The frequency window is determined by the energy range of 1.6–12.4 keV corresponding to the GOES SXR channel of 1–8 Å.

## Data availability

The datasets generated during and/or analysed during the current study are available from the corresponding author upon request. The SDO/AIA data are also available at http://jsoc.stanford.edu/ajax/lookdata.html, EOVSA data from http://ovsa.njit.edu/data-browsing.html, GOES data from https://www.swpc.noaa.gov/products/goes-x-ray-flux, and e-CALLISTO data from http://soleil.i4ds.ch/solarradio/callistoQuicklooks/?date=20170713.

## Code availability

EOVSA data processing tools are included in the open-source SunCASA package (https://github.com/suncasa/suncasa-src), which is based on the CASA (https://casa.nrao.edu). Codes for processing GOES and SDO/AIA data are included in the SolarSoftWare (SSW) repository (https://www.lmsal.com/solarsoft/). E-CALLISTO processing codes are available at https://www.e-callisto.org. The core wavelet analysis software is provided by Torrence and Compo[68] and available at http://atoc.colorado.edu/research/wavelets; the open-source code for fitting the background power spectra and determining the significance levels is available at https://idoc.ias.u-psud.fr/MEDOC/wavelets_tc98. The MHD code is accessible at https://princetonuniversity.github.io/Athena-Cversion/.

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

## Acknowledgements
Y.K., X.C., Y.W. and M.D. are funded by NSFC grants 11722325, 11733003, 11790303 and by National Key R&D Program of China under grant 2021YFA1600504. X.C. is also supported by Alexander von Humboldt foundation. B.C. and S.Y. acknowledge support by US NSF grants AGS-1654382 and AST-2108853 to NJIT. EOVSA operations are supported by US NSF grants AST-1910354 and AGS-2130832 to NJIT.

## Author contributions
X.C. led the project. Y.K. and S.Y. analysed the data. Y.W. performed MHD simulations. X.C. and Y.K. wrote the manuscript. B.C., E.P.K and M.D. took participate in the discussion and revised the manuscript.

## Competing interests
The authors declare no competing interests.
