## [Peer Review File · Nature Communications]

Reviewers' comments:

Reviewer #1 (Remarks to the Author):

The paper presents an interesting case study of the phenomenon of quasi-periodic pulsations (QPP) in the microwave (by EOVSA) and extreme ultraviolet (by SDO/AIA) emissions from a weak solar flare, apparently caused by a quasi-periodic regime of magnetic reconnection in the current sheet region. The manuscript is well written, the figures are well designed, and the results look technically correct (the co-authors are certainly among the world-leaders in the field). However, the work does not seem to be put in a proper context of the previous and current worldwide activity in the QPP research. The novelty element of this work, its breakthrough and transformative nature that would warrant a publication in Nature Comm are not clear to this referee. The authors claim that the observational data used for this study is "unprecedented", while a huge amount of previous relevant works on observations of QPP events in the microwave band with such instruments as e.g. NoRH and NoRP, and the radioheliographs of a new generation -- SRH and MUSER, some of which also possess similarly high spatiotemporal resolution and spectral imaging capabilities, is missing (not even mentioned in the text). The modeling part of the manuscript does not seem to account for the wealth of comprehensive results on oscillatory reconnection, obtained by research teams at e.g. Northumbria University, Ondrejov, KU Leuven, and Kyoto University. The lack of a proper context affects the presentation and may influence the apparent scientific value of this work. Likewise, the authors did not put enough effort into the introduction/discussion of the mechanisms for modulation of the microwave emission from solar flares by MHD waves, proposed by e.g. 2012ApJ...748..140M and 2015SoPh..290.1173K. By the way, simulation results presented in the former work (2012ApJ...748..140M) were already applied to observations from OVSA, which again affects the novelty element of this work. In this respect, the presented manuscript seems to be more suitable for a more specialized journal (e.g. ApJ, A&A, MNRAS, or Sol Phys) after addressing the above issues.

Reviewer #3 (Remarks to the Author):

The article demonstrates the presence of quasi-periodic pulsations in a flare in microwave and the derivative of SXR. Comparing with EUV observations from AIA, the authors associate the microwaves with two different sources: the flaring loops and the current sheet region. MHD simulations produce a similar signature in a synthetic SXR light curve. Combining the simulations and the observational results, the authors conclude that the QPPs are driven by quasi-periodic flare magnetic reconnection, which arise from the modulation of magnetic islands within the current sheet.

The results are definitely interesting and worthy of publication somewhere but I feel that the authors should better describe the novelty of the results in order to justify publication in this particular journal. After all, this is not the first time QPP have been observed in these wavebands for a single flare. A more systematic review of the current literature that describes more clearly the significance of these results would be appreciated.

The analysis itself is predominantly good – I just a few minor queries listed below.

P5: 'We find that the speed varies in time with the [GOES SXR flux]' – could you clarify this? Do you mean it varies in phase is the SXR flux?

P7: Please provide further details of the wavelet transform as there are many variants of this method (for example, what mother wavelet is used). Please clarify how the significance levels were determined? From the caption of Fig. 3: 'Note that a white noise is adopted...' – I am not sure what you mean by this – is this for determining the significance levels? Are you using the standard method of Torrence and Compo? Often red noise is a better assumption – could you justify the assumption of white noise if that is what you mean. The work of <https://ui.adsabs.harvard.edu/abs/2016ApJ...825..110A/abstract> provides suggestions of how to improve on the standard Torrence and Compo significance levels – this work would be more rigorous if this approach were incorporated – this will be particularly important when assessing the significance of the period at 16s, which is clearly less significant.

Also regarding Fig 3, and more specifically the detrending – why was a 61s window chosen and how can periodicity of ~30s remain, when the smoothing window is 30s as is implied by the last sentence.

For Figure 4d, what are the dashed lines?

There is a lot of discussion over how much agreement there is in terms of the QPPs observed but it is all qualitative and vague – e.g. looks similar. Could more precise information be obtained by performing a correlation analysis?

Please double check the values in Section III – this is not my area of expertise but as far as I can tell the characteristic mass density should be $40^2/(4.9e7)^2$, (as in cgs units magnetic permeability is 1?) which does not equal the value given of $5.31e-14g/cm^3$. This then has a knock-on effect on other values quoted which are based on ρ_0 . For example, according to the text, $p_0 = \rho_0 * u_0^2$, but $\rho_0 = \mu_0 * B_0^2 / u_0^2$, and so $p_0 = B_0^2 = 160Pa$?

Reply to Referees

We are very grateful to your valuable criticisms and suggestions. Based on your criticisms and suggestions, we re-organized our Introduction and extended Discussion to emphasize that this work provides the most convincing evidence for flare QPPs arising from the oscillatory magnetic reconnection process within the flare current sheet. In order to more suitably specify our new results, we also change the title of the paper to “Microwave Imaging of Quasi-periodic Pulsations at Flare Current Sheet”. A detailed reply to all your comments is also listed below.

Reviewer #1 (Remarks to the Author):

The paper presents an interesting case study of the phenomenon of quasi-periodic pulsations (QPP) in the microwave (by EOVS) and extreme ultraviolet (by SDO/AIA) emissions from a weak solar flare, apparently caused by a quasi-periodic regime of magnetic reconnection in the current sheet region. The manuscript is well written, the figures are well designed, and the results look technically correct (the co-authors are certainly among the world-leaders in the field). However, the work does not seem to be put in a proper context of the previous and current worldwide activity in the QPP research. The novelty element of this work, its breakthrough and transformative nature that would warrant a publication in Nature Comm are not clear to this referee.

R: Firstly, we sincerely thank you for your careful review and constructive comments. In order to more distinctly emphasize the novelty and breakthrough of our work, we re-organize *Introduction* and extend *Discussion* section in the context of the QPP research field. We find that, although previous publications took advantage of EUV and microwave imaging data, but they only imposed some restrictions on diagnosing the QPP mechanism, the direct evidence for QPPs at the current sheet caused by the oscillatory reconnection within has been absent. As far as we know, it is for the first time that we locate the QPP sources at the current sheet of solar flares, which strongly supports the interpretation of quasi-periodic magnetic reconnection. Such statements can be found at the *Introduction* and *Discussion* in the new manuscript.

The authors claim that the observational data used for this study is “unprecedented”, while a huge amount of previous relevant works on observations of QPP events in the microwave band with such instruments as e.g. NoRH and NoRP, and the radioheliographs of a new generation -- SRH and MUSER, some of which also possess similarly high spatiotemporal resolution and spectral imaging capabilities, is missing (not even mentioned in the text).

R: Thanks for your comments. The new *Introduction* includes more previous imaging observations on QPPs at microwave bands, mostly led by NoRH (and sometimes with NoRP), see the fourth paragraph in *Introduction*. We agree that NoRH has a high spatio-temporal resolution. However, it only provides images at 17 and 34 GHz. NoRP only provides the spatially-integrated spectra. By contrast, at the time of the observation, EOVS is able to image at up to 30 spectral windows from the relatively low frequency 3.4 GHz to 17.9 GHz, and therefore the spatially-resolved spectra map. The low frequency bands were shown to be very critical for spatially resolving the structure above flare loops including the key reconnection current sheet. Recent results have clearly shown that some extended microwave sources exist along the reconnection current sheet (see Chen et al. 2021, doi: 10.1038/s41550-020-1147-7).

With regard to the new-generation radioheliographs SRH, as far as we know, SRH ever observed QPPs, but which are found to be located at the flare region not along the CS. In fact, for such a disk event, it is almost impossible to detect the current sheet structure and thus the oscillatory magnetic reconnection process.

The second author is one of members of MUSER scientific committee. At present, MUSER only provides observations below 2 GHz (down to 400 MHz), which are not the best bands for detecting the current sheet. The high frequency array (2-15 GHz) of MUSER is still under calibration and no scientific data are available at the moment.

The modeling part of the manuscript does not seem to account for the wealth of comprehensive results on oscillatory reconnection, obtained by research teams at e.g. Northumbria University, Ondrejov, KU Leuven, and Kyoto University.

R: The purpose of running an MHD simulation is to synthesize the SXR emissions and compare with observations. Here, we do not focus on the advance of our MHD results, the discussion of which can be found in our recent two publications (Wang et al. 2021, 2022; doi: 10.3847/1538-4357/ac3142 and 10.3847/2041-8213/ac715a). Nevertheless, to address this concern you proposed, in the revised version, we newly added some discussions and comparisons with the similar simulation results in Section III as “In fact, the quasi-periodic characteristics related to the generation of island in the CS has been revealed by numerous previous MHD simulations\cite{kliem2000,barta2008,bhattacharjee2009,huang2010,guo2013,guidoni2016,zhao2020}. In particular, it is pointed out that waves and shocks generated near the ends of CS by (even quasi-steady) reconnection outflows could be oscillatory and responsible for QPPs\cite{takasao2016,takahashi2017}. Additionally, the quasi-periodic deformation of current sheets near magnetic null points might also correspond to QPPs\cite{murray2009,mclaughlin2009,mclaughlin2012,thurgood2017}.”

The lack of a proper context affects the presentation and may influence the apparent scientific value of this work. Likewise, the authors did not put enough effort into the introduction/discussion of the mechanisms for modulation of the microwave emission from solar flares by MHD waves, proposed by e.g. 2012ApJ...748..140M and 2015SoPh..290.1173K. By the way, simulation results presented in the former work (2012ApJ...748..140M) were already applied to observations from OVSA, which again affects the novelty element of this work.

R: Thanks for your suggestions. We have incorporated these previous publications into our new manuscript. However, we still want to address that, in these papers, the MHD waves are just used for interpreting the QPPs as observed at the flare loops, and the direct evidence for MHD waves modulating the parameters of flare loops is not found in our work. While, with the combination of EUV images, we directly observe the sources of QPPs at the current sheet at different frequencies, which are more inclined to support the oscillatory reconnection model. Such statements can be found at the third paragraph of *Summary and Discussion* as “The periodicity of the QPPs we derived seems to support the interpretation of the sausage mode ... is essentially equivalent to external MHD waves giving rise to the quasi-periodic reconnection within the CS.”

In this respect, the presented manuscript seems to be more suitable for a more specialized journal (e.g. ApJ, A&A,

MNRAS, or Sol Phys) after addressing the above issues.

R: We really thank your comments and suggestions and hope our significant updates in the new version can address your main concerns.

Reviewer #3 (Remarks to the Author):

The article demonstrates the presence of quasi-periodic pulsations in a flare in microwave and the derivative of SXR. Comparing with EUV observations from AIA, the authors associate the microwaves with two different sources: the flaring loops and the current sheet region. MHD simulations produce a similar signature in a synthetic SXR light curve. Combining the simulations and the observational results, the authors conclude that the QPPs are driven by quasi-periodic flare magnetic reconnection, which arise from the modulation of magnetic islands within the current sheet.

The results are definitely interesting and worthy of publication somewhere but I feel that the authors should better describe the novelty of the results in order to justify publication in this particular journal. After all, this is not the first time QPP have been observed in these wavebands for a single flare. A more systematic review of the current literature that describes more clearly the significance of these results would be appreciated.

R: Thanks very much for your positive comments and suggestions. In the revised version, we re-organize Introduction in the context of the QPP research field to distinctly emphasize the novelty and breakthrough of our work. Also see the reply to the similar concern raised by Referee 1.

The analysis itself is predominantly good – I just a few minor queries listed below.

P5: ‘We find that the speed varies in time with the [GOES SXR flux]’ – could you clarify this? Do you mean it varies in phase with the SXR flux?

R: Thanks for your correction, we have updated the sentence as “it varies *in phase* with the GOES SXR flux.”

P7: Please provide further details of the wavelet transform as there are many variants of this method (for example, what mother wavelet is used). Please clarify how the significance levels were determined? From the caption of Fig. 3: ‘Note that a white noise is adopted...’ – I am not sure what you mean by this – is this for determining the significance levels? Are you using the standard method of Torrence and Compo? Often red noise is a better assumption – could you justify the assumption of white noise if that is what you mean. The work of <https://ui.adsabs.harvard.edu/abs/2016ApJ...825..110A/abstract> provides suggestions of how to improve on the standard Torrence and Compo significance levels – this work would be more rigorous if this approach were incorporated – this will be particularly important when assessing the significance of the period at 16s, which is clearly less significant.

R: (1) Clarification: we used the standard wavelet procedure provided by Torrence and Compo, the mother wavelet used is ‘MORLET’, and the white noise is adopted to determining the significance levels. (2) The main reason why we used the white noise instead of red noise is for the simplicity at the situation that the oscillatory behavior of the lightcurves is obvious and the resultant periods can be visually checked in the lightcurves. (3) We have

incorporated the method you suggested in the revised manuscript, which fits the background spectra with a combination of power-law function and a constant value and determines the 95% *global* significance level to more confidently confirm the periodicity.

Detailed modifications can be found in the second paragraph of Section II.C as highlighted in bold and in Figure 3.

Also regarding Fig 3, and more specifically the detrending – why was a 61s window chosen and how can periodicities of ~30s remain, when the smoothing window is 30s as is implied by the last sentence.

R: In the new manuscript, we change to use the new method you suggested to determine the significance levels. The detrending is not needed any more and the previous sentence has been deleted.

For Figure 4d, what are the dashed lines?

R: Same as in Figure 3. New descriptions have been added.

There is a lot of discussion over how much agreement there is in terms of the QPPs observed but it is all qualitative and vague – e.g. looks similar. Could more precise information be obtained by performing a correlation analysis?

R: In appendix, we added the cross-correlation analyses between the lightcurves of microwave emissions at the CS region at 8.4 GHz and the SXR flux derivative, as well as that among the lightcurves and temporal variations of three spectral indices.

Please double check the values in Section III – this is not my area of expertise but as far as I can tell the characteristic mass density should be $40^2/(4.9e7)^2$, (as in cgs units magnetic permeability is 1?) which does not equal the value given of $5.31e-14\text{g/cm}^3$. This then has a knock-on effect on other values quoted which are based on ρ_0 . For example, according to the text, $p_0 = \rho_0 u_0^2$, but $\rho_0 = \mu_0 B_0^2 / u_0^2$, and so $p_0 = B_0^2 = 160\text{Pa}$?

R: We have rechecked all values and confirmed that all values listed in this section are correct, but our expressions may cause a confusion between the S.I. and c.g.s. units. Actually, these units are deduced under S.I. unit system which are then converted into cgs-type. The formula of Alfvén speed in S.I. system is $u_0 = B_0 / \sqrt{\mu_0 \rho_0}$, while in c.g.s. system is $u_0 = B_0 / \sqrt{4\pi \rho_0}$. Therefore, if deduced under c.g.s. system, $\rho_0 = B_0^2 / (4\pi u_0^2) = 40^2 / (4\pi * 4.9^2) \times 10^{-14} = 5.31 \times 10^{-14}$, which is consistent with our values. The unit of pressure is now $p_0 = B_0^2 / 4\pi = 127.3 \text{ dyn/cm}^2 = 12.73 \text{ Pa}$. To avoid the confusion, in the revised manuscript, we unify the expression under c.g.s. systems.

REVIEWERS' COMMENTS

Reviewer #1 (Remarks to the Author):

The authors have indeed managed to improve the placing of their results into the context of other works on the phenomenon of QPP in microwave observations of solar flares and other works on their modelling in terms of oscillatory reconnection. I also appreciate their attempt to point out the apparently first direct evidence of QPP in a current sheet as the novel element of this work. However, I still cannot recommend it for publication in Nature Comm, as in my opinion the presented advances are far too mainstream and would therefore be more suitable for a more specialised journal as suggested in my previous report.

I would also like to attract the authors' and editors' attention to the coherency of opinions of both referees in the previous round of review.

Reviewer #3 (Remarks to the Author):

I thank the authors for making the modifications. The case for the novelty and importance of the results is now much clearer and as a result, I am happy for this work to be published.

Response to Reviewers:

Reviewer #1 (Remarks to the Author):

The authors have indeed managed to improve the placing of their results into the context of other works on the phenomenon of QPP in microwave observations of solar flares and other works on their modelling in terms of oscillatory reconnection. I also appreciate their attempt to point out the apparently first direct evidence of QPP in a current sheet as the novel element of this work.

R: We appreciate the reviewer for his/her constructive suggestions and comments in the last round of review, which significantly helps improve the manuscript.

However, I still cannot recommend it for publication in Nature Comm, as in my opinion the presented advances are far too mainstream and would therefore be more suitable for a more specialised journal as suggested in my previous report.

I would also like to attract the authors' and editors' attention to the coherency of opinions of both referees in the previous round of review.

R: Though the reconnection model being one of prevalent explanations for solar flare QPPs, for a long time there is no direct observational evidence. We believe that QPPs well imaged at the flare current sheet provide a strong support for the reconnection interpretation.

Reviewer #3 (Remarks to the Author):

I thank the authors for making the modifications. The case for the novelty and importance of the results is now much clearer and as a result, I am happy for this work to be published.

R: We thank the reviewer for his/her positive attitude and useful suggestions on the improvement of the manuscript.